# Effects of massive transfusion (10-20 litres) versus ultramassive transfusion (≥20 litres) on mortality in adult liver transplant recipients: A propensity-score matched study

Zac Tran[1☯], Nattaya Raykateeraroj[2☯], Jemin Suh[1], Jordan Ismail[1], Michael Fink[3,4], Rebecca Caragata[1], Marcos V. Perini[3,4], Anoop N. Koshy[5], Dong-Kyu Lee[6], Laurence Weinberg[1,7]*

1 Department of Anaesthesia, Austin Health, Heidelberg, Victoria, Australia, 2 Department of Anesthesiology, Faculty of Medicine Siriraj Hospital, Mahidol University, Bangkok Noi, Bangkok, Thailand, 3 Department of Surgery (Austin Precinct), The University of Melbourne, Melbourne, Victoria, Australia, 4 Victorian Liver Transplant Unit, Austin Health, Melbourne, Victoria, Australia, 5 Department of Cardiology, Austin Health, Heidelberg, Victoria, Australia, 6 Department of Anaesthesiology and Pain Medicine, Dongguk University Ilsan Hospital, Goyang, the Republic of Korea, 7 Department of Critical Care, The University of Melbourne, Melbourne, Victoria, Australia

☯ These authors contributed equally and co-first authors to this work.
* laurence.weinberg@austin.org.au

## Abstract

### Background

Ultramassive perioperative fluid transfusion during orthotopic liver transplantation (OLT) identifies a high-risk recipient phenotype and is associated with substantially increased mortality compared with conventional massive transfusion. OLT frequently necessitates high-volume fluid resuscitation, yet the prognostic implications of ultramassive perioperative transfusion in this setting remain uncertain. We evaluated whether ultramassive transfusion (≥20 L total perioperative fluid) is independently associated with worse survival than massive transfusion (10–<20 L) in adult OLT recipients.

### Methods

In this single-centre retrospective cohort (2009–2023), we included adults undergoing primary OLT who received ≥10 L of total perioperative fluid (crystalloids, colloids, blood, and blood products administered intraoperatively and within 24 hours postoperatively). Propensity score matching was used to compare recipients receiving ultramassive (≥20 L) versus massive (10–<20 L) transfusion, balancing recipient, donor, and intraoperative characteristics. Primary outcomes were 90-day, 3-year, and overall all-cause mortality; secondary outcomes included graft failure, graft and patient survival time, primary non-function, early allograft dysfunction, thrombotic complications, acute kidney injury, and hospital length of stay.

**Data availability statement:** All relevant data are within the paper and its Supporting Information files.

**Funding:** The author(s) received no specific funding for this work.

**Competing interests:** The authors have declared that no competing interests exist.

## Results

Of 993 OLT recipients, 306 (30.8%) received ≥10 L perioperative fluid and comprised the unmatched cohort. In the propensity-matched cohort (n = 188), ultramassive transfusion was associated with significantly higher 90-day, 3-year and overall mortality compared with massive transfusion, with consistent effect estimates across multiple matching strategies and sensitivity analyses. In contrast, ultramassive transfusion showed no robust or consistent association with early allograft dysfunction, thrombotic complications, graft loss, or other secondary endpoints, likely reflecting low event rates and limited power.

## Conclusions

Among adult OLT recipients exposed to high-volume perioperative resuscitation, ultramassive transfusion delineates a distinct high-risk phenotype characterised by substantially increased short- and long-term mortality. Although causal inference is precluded by the observational design, these findings suggest that reaching an ultramassive transfusion threshold may mark a particularly vulnerable subgroup that warrants intensified intraoperative optimisation strategies and tailored long-term surveillance.

## Introduction

Orthotopic liver transplantation (OLT) is the definitive intervention for end-stage liver disease, but it remains a complex and resource-intensive surgery with a high risk of intraoperative haemorrhage due to hepatic vascularity, portal hypertension, and disease-related coagulopathy [1–5]. Consequently, high-volume transfusions are often necessary to maintain haemodynamic stability and perfusion [6,7]. Massive fluid transfusion (MT), defined herein as a total perioperative fluid volume of ≥10 litres administered intraoperatively and within 24 hours postoperatively, is a recognised risk factor for adverse surgical outcomes, including elevated mortality, multiorgan failure, and resource consumption [8–10].

Within this MT population, a higher-risk subgroup of ultramassive fluid transfusions (UMT) exists, defined herein as a total perioperative fluid volume of ≥20 litres administered intraoperatively and within 24 hours postoperatively. The incidence of UMT transfusions is increasing among OLT recipients, which may represent a clinical and logistical inflection point for disproportionately increased resource consumption and worse outcomes [11–13]. Unfortunately, the evidence is hindered by several factors: existing massive transfusion protocols are predominantly trauma-based and may not translate to the pathophysiology of OLT recipients who are predisposed to severe haemorrhage and physiological derangement [14]; studies do not rigorously account for confounding, such that differences in outcomes may be driven by underlying morbidity rather than the transfusion volume [11,15]; few studies explore other clinically relevant outcomes such as graft dysfunction, thrombotic events or length

of stay [10,16]; and the absence of a consensus definition of high-volume transfusion thresholds complicates cross-study comparison [10,11,17].

Given the cost, scarcity and logistical burden of blood products, understanding the clinical impact of UMT in OLT is critical for prognostication, resource stewardship and designing OLT-specific transfusion protocols [11,16–21]. There is a gap in the literature for outcome-based transfusion studies in non-trauma contexts, especially in OLT, where the effects of high-volume transfusion remain poorly characterised [13,16,22] and existing protocols may be inadequate. To our knowledge, few studies have used propensity score matching (PSM) to rigorously compare the postoperative outcomes of UMT versus MT in the OLT setting [11,12,16,23].

This single-centre retrospective cohort study aimed to compare postoperative outcomes of OLT recipients who received UMT versus MT using PSM. As there is no universally accepted definition of high-volume transfusion, particularly in the context of orthotopic liver transplantation, we defined total perioperative fluid volume *a priori* as the sum of all crystalloids, colloids, and blood products administered intraoperatively and within 24 hours postoperatively, on the basis that all fluids contribute to the physiological derangements associated with high-volume resuscitation [10–12,14,16]. The primary outcome was patient survival during the early (90 days), intermediate (3 years) and long-term postoperative periods. Secondary outcomes included graft loss across the same postoperative periods, as well as exploratory outcomes of long-term transplant-associated complications, acute kidney injury, thrombotic events, and hospital length of stay. Based on existing literature suggesting a dose-dependent effect of transfusion volume on adverse outcomes, we hypothesised that UMT would be associated with significantly poorer outcomes than MT, even after adjustment for confounding [8,10,14].

## Methods

### Ethics

The Austin Health Human Research Ethics Committee (HREC/105884/Austin-2024) approved this study (no: 12624000499583) on 25 March 2024 and waived the requirement for informed consent due to the retrospective design and use of de-identified, routinely collected clinical data. The study adhered to the principles of the Declaration of Helsinki and the Declaration of Istanbul and was reported in accordance with the STROCSS 2025 guidelines. Data collection was completed on 4 March 2025. No protocol changes occurred after ethics approval. Data analysis commenced only after approval was granted.

### Study design and setting

This is a retrospective cohort study of adult OLT recipients (≥18 years old) at Austin Health, Melbourne, Australia, a quaternary referral centre and home to the Victorian Liver Transplant Unit, from 14 November 2009–31 December 2023. The unit performs approximately 100 adult deceased donor transplants annually.

### Patient population and definitions

All adult OLT recipients who received ≥10 L of total fluids intraoperatively and within 24 hours postoperatively were eligible, excluding those with combined cardiac-liver or lung-liver co-transplantations. Patients were not excluded based on transplant indication or re-transplantation status; repeat transplantations were treated as separate events. The unmatched cohort included all eligible participants, who were stratified according to the total fluid volume administered intraoperatively and within 24 hours postoperatively. Massive transfusion was defined as administration of 10–20 L of total fluid, and ultramassive transfusion (UMT) as ≥20 L of total fluid, encompassing crystalloids, colloids, and blood products (packed red blood cells [pRBC], fresh frozen plasma [FFP], and platelets). This composite definition captures the total perioperative resuscitation burden, reflecting the physiological impact of extreme fluid transfusion, and is consistent with prior work in this field, including a recent study of ultramassive fluid transfusions at our institution [24]. Although not universally

accepted, this approach is consistent with recent literature and represents clinically meaningful checkpoints for resource consumption and outcomes [11,17]. Follow-up was conducted longitudinally through the institutional transplant registry, from the index procedure until graft loss, mortality, or the end of the study period.

Our comparison of UMT and MT recipients evaluates the incremental risk associated with extreme transfusion volumes in the high-acuity OLT setting. Patients who received <10 L were excluded, as this group is often heterogeneous with potentially lower acuity [10,14], which could confound the relationship between transfusion volume and outcomes. The 10 L threshold is a widely accepted, if not universal, definition of massive transfusion and is commonly used to trigger clinical protocols and heighten vigilance [11,14,25]. Restricting the cohort to recipients who received ≥10 L allowed for a more focused evaluation of whether transfusion volumes exceeding this threshold confer disproportionately worse outcomes and warrant more vigilant perioperative care.

## Outcomes

The primary outcomes were 90-day, 3-year and overall mortality, recorded cumulatively from the index liver transplantation until the last follow-up, as there was no fixed follow-up period.

Secondary outcomes were 90-day, 3-year and overall graft failure (defined as re-transplantation or death attributed to graft failure occurring at any time post-transplant); patient and graft survival time (from the index procedure to death or graft failure, respectively); primary non-function (PNF, defined as re-transplantation or patient death due to graft failure within 7 days); early allograft dysfunction (EAD, defined as aspartate transaminase (AST) or alanine transaminase (ALT) >2000 IU/L at least once within the first 7 postoperative days (absolute value); bilirubin ≥10 mg/dL or international normalized ratio [INR] ≥1.6 within 7 days); thrombotic complications (hepatic artery, portal vein, hepatic vein, or other sites); acute kidney injury (AKI) and stage (per Kidney Disease: Improving Global Outcomes [KDIGO] criteria); and hospital length of stay (LOS, total postoperative days). All outcomes were obtained from the institutional liver transplant registry and verified by electronic medical record review to ensure accuracy. Events were recorded cumulatively from the date of transplantation until the end of data collection.

## Data collection

A comprehensive dataset of donor, recipient, intraoperative, and postoperative variables was extracted from the institutional liver transplant registry and supplemented by manual review of electronic medical records using a standardised abstraction protocol. The registry data comprised information from routine clinical assessments and laboratory investigations.

Donor variables included age, baseline laboratory values, donation pathway (donation after cardiac death or brain death), donor risk index (DRI), cause of death, cold ischaemia time (time in preservation solution), warm ischaemia time (time from removal from cold storage to reperfusion), total ischaemia time, and partial graft status.

Recipient variables included age, sex, height, weight, body mass index (BMI), transplant indication, model for end-stage liver disease scores (MELD, MELD-sodium, and MELD-3), Gender-Equity Model for Liver Allocation (GEMA), and baseline laboratory values (coagulation profile, liver function, renal function, full blood count, and electrolytes).

Intraoperative variables included concurrent organ transplants, the volume of administered blood products and fluids (albumin, plasmalyte, pRBC, FFP, cryoprecipitate, platelets, autologous wash cells, and organ donor wash cells), and total fluid volume.

## Statistical analyses

Statistical analyses were performed using R v4.5.0 [26], with the packages *MatchIt* (4.7.1) [27] for PSM, and *survival* (3.8.3) [28] and *coxphf* (1.13.4) [29] for survival analysis. All data were thoroughly assessed for normality, outliers, and missing values. Outliers, identified visually and statistically, were retained following reconciliation with electronic health records,

acknowledging the expected clinical extremes within this cohort. Normality was evaluated using the Shapiro-Wilk test and Q-Q plots, with non-parametric tests applied when assumptions were violated. Variables with less than 5% missingness were considered to have minimal missing data. For variables with greater than 5% missingness, Little's MCAR test and comparative profiling suggested that the data were unlikely to be missing completely at random and were more consistent with a missing at random mechanism. Given the minimal missingness in key propensity score and outcome variables, and the availability of composite scores such as MELD and DRI that captured several laboratory components, a complete-case analysis was performed (S1 Table). Multiple imputation was not performed, as missingness in the primary analytic variables was limited and imputation of secondary laboratory variables was unlikely to materially affect the primary analysis.

After evaluating multiple matching configurations, an optimal 1:1 strategy without replacement was used to match UMT and MT recipients. Propensity for UMT was estimated using 11 clinically relevant preoperative variables: age, sex, BMI, transplant indication, MELD-3 score, baseline albumin, baseline platelets, DRI, donation pathway, cold ischaemia time, and partial graft transplantation. Covariate balance before and after matching was assessed by comparing standardised mean differences (SMD), visualised via Love plots, with an SMD < 0.1 indicating adequate balance..

Continuous variables are presented as mean ± standard deviation or median [interquartile range], based on their distribution. Categorical variables are reported as frequencies and percentages. For unmatched group comparisons, the Student's t-test or the Mann-Whitney U test was used for continuous variables, and the chi-square test or Fisher's exact test for categorical variables. In the matched cohort, the paired t-test or Wilcoxon signed-rank test was applied for continuous variables, while McNemar's test was used for binary variables. Logistic regression was used for binary outcomes in the unmatched cohort, and conditional logistic regression stratified by matched pair was used for the matched cohort. When conditional logistic regression was infeasible due to sparse data or quasi-complete separation, non-parametric tests were utilised; if statistical comparison remained infeasible, descriptive statistics were reported.

Survival analyses used Kaplan–Meier curves with log-rank tests and Cox proportional hazards regression. A standard Cox model was fitted in the unmatched cohort, and a conditional Cox model stratified by matched pair was used in the matched cohort. Mortality was analysed with patient death as the event of interest. Graft loss was analysed separately using a cause-specific Cox model in which deaths without prior graft failure were censored, yielding hazard ratios that are directly comparable to the primary mortality analysis; formal competing-risks regression was not undertaken because of the small number of graft-failure events and the exploratory nature of the study. The proportional hazards assumption was assessed using Schoenfeld residuals.

At 90 days, zero mortality events in the MT group resulted in complete separation, necessitating Firth's penalised Cox and logistic regression to obtain estimates. This approach was applied only where standard models failed to converge. Formal assessment of the proportional hazards assumption was not performed for these models, as Schoenfeld residuals are not defined for Firth's penalised likelihood. This approach preserves the time-to-event framework in the primary analysis.

Three sensitivity analyses were conducted to assess the robustness of the primary findings. First, UMT recipients were compared with all non-UMT recipients (<20 L of total fluids) using a 1:3 nearest-neighbour propensity-score matching approach. Covariate balance was assessed using SMDs.

Second, a pRBC-centred sensitivity analysis was conducted to address potential conflation of physiologically distinct fluid types in the primary analysis. Intraoperative pRBC units transfused were used to define the exposure groups: an MT group (10–15 units) and a UMT group (≥15 units). The lower MT boundary (≥10 units) is consistent with the literature. While a UMT threshold of ≥20 units was initially considered, a threshold of ≥15 units was adopted to ensure sufficient sample size for matching and subsequent analysis. A nearest-neighbour 1:1 strategy with a caliper of 0.2 standard deviations was used. Covariate balance was assessed using SMDs.

Third, a time-period sensitivity analysis was performed to assess whether secular changes in clinical practice over the 15-year study period could account for the observed association between UMT and patient survival in the primary analysis. The median transplant date (29 October 2018) was used to define two temporal periods: an early era (December

2009 to October 2018) and a contemporary era (October 2018 to December 2023). Rather than splitting the matched cohort by era, which would separate matched pairs transplanted in different eras, the transplant era was incorporated as a covariate within the existing conditional Cox models in the primary analysis. Era-adjusted models were fitted to test whether the UMT-mortality association persisted after accounting for the transplant period. Additionally, models fitted with an interaction term (UMT × era) were used to assess whether the magnitude of the UMT effect differed between eras. The 90-day patient survival was not modelled due to complete separation within eras, rendering Cox models not estimable.

A two-tailed $p$-value < 0.05 was considered significant. Effect sizes were estimated and are presented with 95% confidence intervals.

## Results

### Participants

In total, 993 patients received a transfusion during their procedure over the study period, of which 93 were excluded as they were combined cardiac-liver or lung-liver co-transplants or paediatric transplants. Of 900 eligible cases screened, 594 were excluded for receiving <10 L of intraoperative fluids within 24 hours, leaving 306 eligible patients for the unmatched analysis: 95 in the UMT group and 211 in the MT group. Subsequently, optimal 1:1 matching without replacement yielded 94 well-balanced matched pairs ($n = 188$), with 118 patients excluded from the matched cohort. Fig 1 presents a detailed flow diagram of participant inclusion and exclusion.

### Baseline characteristics

Table 1 presents the baseline demographic, clinical, laboratory, and donor characteristics for the unmatched cohort ($n = 306$) and the matched cohort ($n = 188$). The complete list of laboratory and donor variables is available in S2 Table. In the unmatched cohort, the UMT and MT groups were comparable across most recipient and donor variables, including recipient age (median 58.2 vs 56.6 years, $p = 0.604$), sex (66.3% vs 73.0% male, $p = 0.276$), and MELD score (median 19.0 vs 21.0, $p = 0.226$). Only three variables showed statistically significant differences: baseline ALT (median 32.0 vs. 41.0 IU/L, $p = 0.039$), cold ischaemia time (median 376.5 vs. 331.0 min, $p = 0.032$), and total ischaemia time (median 431.0 vs. 376.5 min, $p = 0.008$).

Various matching strategies utilising alternative ratios, calipers and algorithms were evaluated; an optimal 1:1 strategy provided the strongest overall balance according to the Love plots of SMDs after matching (S1 Fig; S3 Table). After propensity score matching, all covariates achieved adequate balance (Fig 2). Matched groups showed no residual imbalance in key clinical variables, such as MELD score, ascites, and donor pathway. Although donor GGT remained statistically different, this was considered clinically negligible given the absence of differences in other liver enzyme measures, indicating satisfactory overall matching quality.

As expected, UMT recipients received significantly greater volumes of blood products and fluids in both the matched and unmatched cohorts (all $p < 0.001$), except for washed donor blood, for which both cohorts received negligible volumes (unmatched $p = 0.664$; matched $p = 0.407$). In the matched cohort, UMT recipients consistently received higher volumes of blood products, including pRBC, FFP, platelets, albumin, and crystalloids. The median total fluid volume was 27.2 L vs 13.1 L after matching, confirming a clear exposure contrast between groups.

### Patient survival

In the matched cohort, UMT recipients experienced markedly reduced survival rates across all measured time points compared to MT recipients (Table 2). At 90 days, mortality was 11.7% in the UMT group and 0% in the MT group. This survival difference persisted at 3 years (20.2% vs. 7.4%; OR 2.71, 95% CI 1.14–6.46, $p = 0.024$) and throughout the follow-up period (27.7% vs. 14.9%; OR 2.09, 95% CI 1.02–4.29, $p = 0.044$). Kaplan-Meier (KM) curves (Fig 3; S2 Fig) demonstrated separation early after transplantation, which widened over time (90-day log-rank $p < 0.001$; 3-year $p = 0.010$; and overall $p = 0.012$). In the conditional Cox regression (Table 3), UMT exposure was associated with a significantly elevated risk

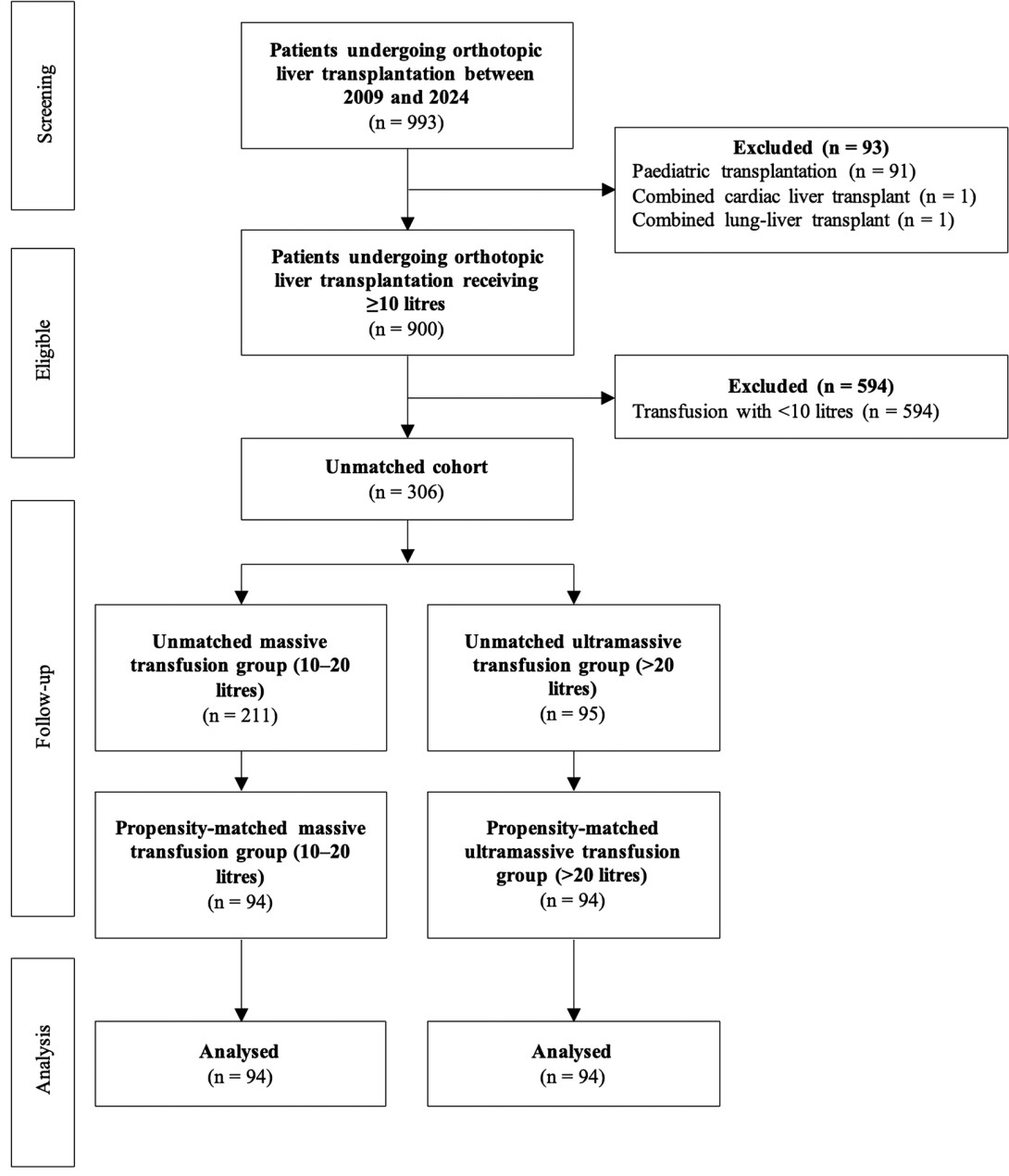

**Fig 1. Flow diagram of participant inclusion and exclusion.**

of mortality after 3 years (HR 2.57, 95% CI 1.07–6.16; $p = 0.034$), which persisted throughout the follow-up period (HR 2.56, 95% CI 1.18–5.52; $p = 0.017$). Complete separation precluded convergence of the conditional Cox model at 90 days; Firth's penalised logistic and Cox regression was therefore applied, yielding statistically significant odds (OR 26.03, 95% CI 3.31–3359.69, p < 0.001; Table 2) and hazard (HR 24.30, 95% CI 3.17–3,119, p < 0.001; Table 3) of mortality at 90 days in UMT recipients. The wide confidence intervals likely reflect the sparse events at 90 days. The Firth model could not be stratified by matched pair and is interpreted as supplementary evidence supporting the primary survival analyses.

**Table 1. Core baseline recipient, donor and intraoperative characteristics before and after matching.**

| Variable | Unmatched ($n=306$) | | | | Matched ($n=188$) | | | |
|---|---|---|---|---|---|---|---|---|
| | UMT | MT | p | SMD | UMT | MT | p | SMD |
| Age (years) | 58.2 [49.7–63.8] | 56.6 [49.5–63.3] | 0.604 | 0.071 | 57.8 [49.5–63.6] | 56.9 [49.7–62.3] | 0.757 | 0.019 |
| Sex: male, n (%) | 63 (66.3) | 154 (73) | 0.276 | 0.160† | 62 (66) | 65 (69.1) | 0.760 | 0.067 |
| BMI (kg/m²) | 28.8 [24.8–32.7] | 27.6 [24.0–32.3] | 0.364 | 0.146† | 28.8 [24.8–32.8] | 28.1 [24.5–33.6] | 0.871 | 0.038 |
| **Transplantation indication** | | | | | | | | |
| Chronic liver disease, n (%) | 51 (53.7) | 102 (48.3) | 0.326 | 0.084 | 50 (53.2) | 50 (53.2) | 0.913 | 0.000 |
| Cancer | 24 (25.3) | 67 (31.8) | | 0.156† | 24 (25.5) | 27 (28.7) | | 0.073 |
| Acute liver failure | 2 (2.1) | 14 (6.6) | | 0.328† | 2 (2.1) | 3 (3.2) | | 0.074 |
| Metabolic disease | 3 (3.2) | 3 (1.4) | | 0.098 | 3 (3.2) | 2 (2.1) | | 0.061 |
| Other | 11 (11.6) | 17 (8.1) | | 0.135† | 11 (11.7) | 8 (8.5) | | 0.099 |
| Re-transplantation | 4 (4.2) | 8 (3.8) | | 0.065 | 4 (4.3) | 4 (4.3) | | 0.000 |
| **Preoperative clinical and laboratory characteristics** | | | | | | | | |
| MELD-Na | 23.5±8.7 | 24.0±8.3 | 0.628 | – | 23.6±8.7 | 23.3±7.9 | 0.745 | – |
| MELD-3 | 21.8 [15.1–28.9] | 22.3 [16.6–29.0] | 0.453 | 0.060 | 21.9 [15.4–29.0] | 22.3 [15.7–28.1] | 0.754 | 0.021 |
| Baseline albumin (g/L) | 29.0 [26.0–33.0] | 31.0 [26.0–36.0] | 0.234 | 0.144† | 29.0 [26.0–33.0] | 31.0 [26.2–35.8] | 0.448 | 0.100† |
| Baseline platelets (×10⁹/L) | 72.0 [49.0–115.5] | 65.0 [48.0–95.0] | 0.122 | 0.244† | 71.5 [49.0–113.2] | 73.0 [52.2–118.5] | 0.871 | 0.039 |
| **Donor and graft characteristics** | | | | | | | | |
| Donor pathway: DCD, n (%) | 9 (9.5) | 14 (6.6) | 0.482 | 0.126† | 9 (9.6) | 9 (9.6) | >0.999 | 0.000 |
| Donor Risk Index (DRI) | 1.4 [1.3–1.8] | 1.4 [1.1–1.7] | 0.189 | 0.170† | 1.4 [1.3–1.8] | 1.5 [1.2–1.8] | 0.801 | 0.020 |
| Partial graft, n (%) | 4 (4.2) | 11 (5.2) | >0.999 | 0.125† | 3 (3.2) | 4 (4.3) | >0.999 | 0.061 |
| **Intraoperative characteristics** | | | | | | | | |
| Cold ischaemia time (min) | 376.5 [287.5–447.8] | 331.0 [271.8–407.0] | 0.032* | 0.251† | 370.9±110.2 | 368.1±96.1 | 0.855 | 0.025 |
| Warm ischaemia time (min) | 48.0 [40.2–56.8] | 45.0 [40.0–51.2] | 0.051 | – | 48.0 [40.2–56.8] | 47.0 [41.0–53.0] | 0.086 | – |
| Total ischaemia time (min) | 431.0 [332.5–501.5] | 376.5 [316.0–460.0] | 0.008* | – | 425.7±114.7 | 416.0±97.4 | 0.529 | – |
| **Intraoperative transfusion and fluid** | | | | | | | | |
| Albumex (L) | 2.5 [2.1–3.6] | 1.3 [1.1–1.6] | <0.001* | – | 2.5 [2.1–3.6] | 1.3 [1.1–1.6] | <0.001* | – |
| Plasmalyte (L) | 14.0 [11.0–18.5] | 7.0 [6.0–8.0] | <0.001* | – | 14.0 [11.0–18.0] | 7.0 [6.0–8.0] | <0.001* | – |
| Packed red blood cells (unit) | 14.0 [10.7–20.0] | 6.3 [4.0–10.0] | <0.001* | – | 14.0 [10.6–19.5] | 6.8 [4.2–10.0] | <0.001* | – |
| Fresh frozen plasma (unit) | 6.0 [4.0–8.0] | 2.0 [1.0–4.0] | <0.001* | – | 6.0 [4.0–8.0] | 2.0 [1.0–4.0] | <0.001* | – |
| Platelets (unit) | 3.0 [2.0–4.0] | 1.0 [1.0–2.0] | <0.001* | – | 3.0 [2.0–4.0] | 1.5 [1.0–2.0] | <0.001* | – |
| Total volume (L) | 27.4 [22.6–39.1] | 13.2 [11.3–15.5] | <0.001* | – | 27.2 [22.6–38.3] | 13.1 [11.1–15.5] | <0.001* | – |

Continuous variables are presented as mean±standard deviation or median [interquartile range]. Categorical variables are presented as frequencies (percentages). For unmatched comparisons, the independent t-test, the Mann-Whitney U test, the Chi-square test, and Fisher's exact test were used. For matched comparisons, the paired t-test, the Wilcoxon signed-rank test, McNemar's test, McNemar's exact test and the Stuart-Maxwell test were used. Non-parametric tests were used if regression did not converge; descriptive statistics were reported if comparison was infeasible. *$p<0.05$ indicates statistical significance. †SMD>0.1. Abbreviations: ALP, alkaline phosphatase; ALT, alanine aminotransferase; AST, aspartate transaminase; BMI, body mass index; DCD, donation after cardiac death; FMS, fluid management system; GEMA, Gender-Equity Model for Liver Allocation; GGT, gamma-glutamyl transferase; MELD-3, Model for End-Stage Liver Disease 3.0; MELD-Na, Model for End-Stage Liver Disease sodium-corrected variant; MT, massive transfusion; SMD, standardised mean difference; UMT, ultramassive transfusion.

## Graft and other clinical outcomes

Graft loss was analysed using cause-specific models, with death before graft failure treated as a censoring event. The association between UMT and graft loss was less clear than with patient survival. In the matched cohort, UMT recipients exhibited higher overall rates of graft loss compared to MT recipients across all postoperative periods; however, these differences were not significant (Tables 2 and 3). Despite the KM curve for graft survival showing a statistically significant

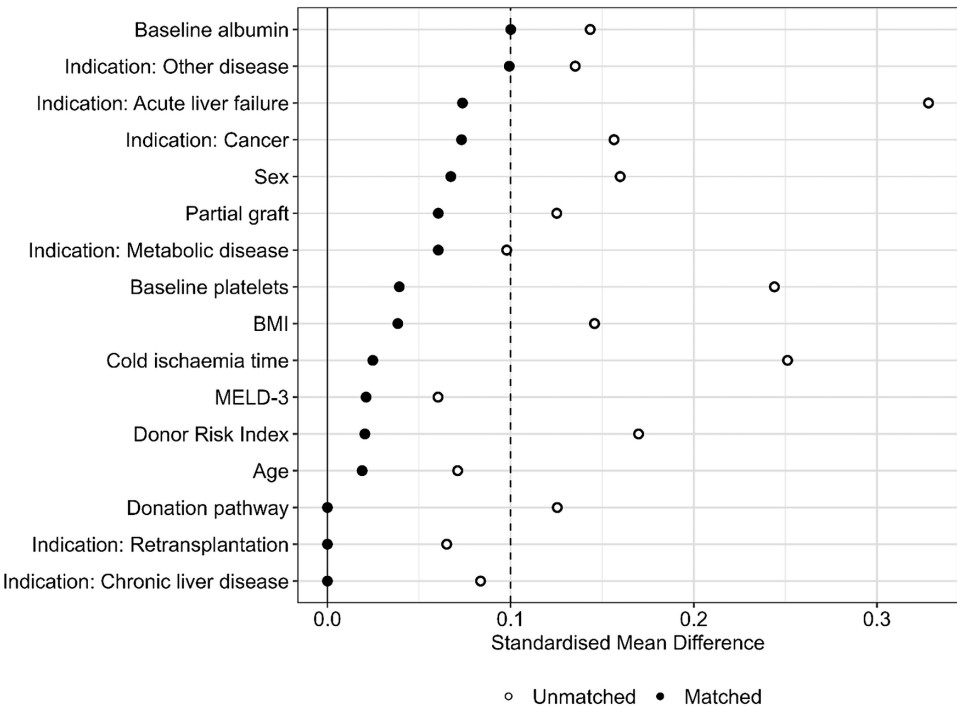

**Fig 2. Love plot of covariate balance after 1:1 optimal matching without replacement.** Standardised mean differences (SMDs) for baseline covariates are compared before and after matching. The vertical dashed line at SMD = 0.1 indicates the threshold for adequate balance.

difference between the groups at 3 years (log-rank $p = 0.037$), this was not consistently corroborated with the KM curves at 90 days (log-rank $p = 0.140$) or overall (log-rank $p = 0.114$) (Fig 4, S2 Fig). While the KM analysis suggests an intermediate-term effect, the conditional logistic and Cox regression models do not corroborate robust evidence of an association between UMT and graft loss at any time point.

The remaining secondary outcomes largely showed no statistically significant difference between the UMT and MT groups in the matched cohort (Table 2). PNF was observed in 5.3% of UMT recipients compared to 1.1% of MT recipients, but this did not reach significance (McNemar's $p = 0.125$). Similarly, there was no significant association between UMT and EAD (43.6% vs. 34.0%, OR 1.41, 95% CI 0.82–2.43, $p = 0.219$). The only thrombotic complication to demonstrate borderline statistical significance was the composite outcome of overall hepatic artery thrombosis (HAT) or portal vein thrombosis (PVT), which was more frequent in the UMT group (8.5% vs. 1.1%, OR 8.00, 95% CI 1.00–63.96, $p = 0.050$). All other individual or composite thrombotic events were not significant. There was no significant difference in the incidence (OR 0.81, 95% CI 0.43–1.53, $p = 0.517$) or severity (OR −0.14, 95% CI −0.40–0.14, $p = 0.294$) of AKI between groups. Although the median LOS was longer for UMT recipients (29.0 days vs. 22.5 days), this was not significant ($p = 0.261$).

The outcomes from the unmatched cohort consistently corroborated the main analysis. Conditional logistic and Cox regression analyses (S4 and S5 Tables) demonstrated significantly higher mortality among UMT recipients across all time points (all $p \leq 0.01$). The KM curves (S3 Fig) likewise demonstrated markedly lower survival probabilities for UMT recipients across 90-day, 3-year, and overall follow-up periods. In contrast, the relationship between UMT and graft survival was not statistically significant at any time point. Unlike in the primary analysis, there was a statistically significant association with EAD, but the association with the composite thrombotic complication of HAT or PVT was not re-established.

**Table 2. Perioperative and long-term outcomes in the propensity-matched cohort.**

| Outcome | Matched (*n* = 188) | | | |
| --- | --- | --- | --- | --- |
| | UMT | MT | Effect size (95% CI) | *p* |
| **Mortality** | | | | |
| 90-day mortality, n (%)† | 11 (11.7) | 0 (0.0) | 26.03 (3.31–3359.69)† | <0.001* |
| 3-year mortality, n (%) | 19 (20.2) | 7 (7.4) | 2.71 (1.14–6.46) | 0.024* |
| Overall mortality, n (%) | 26 (27.7) | 14 (14.9) | 2.09 (1.02–4.29) | 0.044* |
| **Graft outcomes** | | | | |
| PNF, n (%) | 5 (5.3) | 1 (1.1) | — § | 0.125 |
| EAD, n (%) | 41 (43.6) | 32 (34.0) | 1.41 (0.82–2.43) | 0.219 |
| 90-day graft loss, n (%) | 6 (6.4) | 2 (2.1) | 5.00 (0.58–42.80) | 0.142 |
| 3-year graft loss, n (%) | 10 (10.6) | 3 (3.2) | 4.50 (0.97–20.83) | 0.054 |
| Overall graft loss, n (%) | 10 (10.6) | 5 (5.3) | 2.25 (0.69–7.31) | 0.177 |
| **Thrombotic complications** | | | | |
| **Hepatic artery thrombosis (HAT)** | | | | |
| 30-day HAT, n (%) | 1 (1.1) | 0 (0.0) | — § | — § |
| Overall HAT, n (%) | 3 (3.2) | 0 (0.0) | — § | — § |
| **Portal vein thrombosis (PVT)** | | | | |
| 30-day PVT, n (%) | 1 (1.1) | 1 (1.1) | 1.00 (0.06–15.99) | >0.999 |
| Overall PVT, n (%) | 5 (5.3) | 1 (1.1) | 5.00 (0.58–42.80) | 0.142 |
| **Hepatic vein thrombosis (HVT)** | | | | |
| 30-day HVT, n (%) | 0 (0.0) | 0 (0.0) | — § | — § |
| Overall HVT, n (%) | 0 (0.0) | 0 (0.0) | — § | — § |
| **Composite thrombosis** | | | | |
| 30-day HAT or PVT, n (%) | 2 (2.1) | 1 (1.1) | 2.00 (0.18–22.06) | 0.571 |
| Overall HAT or PVT, n (%) | 8 (8.5) | 1 (1.1) | 8.00 (1.00–63.96) | 0.050* |
| 30-day other thrombosis, n (%) | 3 (3.2) | 3 (3.2) | 1.50 (0.25–8.98) | 0.657 |
| Overall other thrombosis, n (%) | 6 (6.4) | 5 (5.3) | 1.20 (0.37–3.93) | 0.763 |
| 30-day any thrombosis, n (%) | 5 (5.3) | 3 (3.2) | 1.67 (0.40–6.97) | 0.484 |
| Overall any thrombosis, n (%) | 13 (13.8) | 6 (6.4) | 2.17 (0.82–5.70) | 0.117 |
| **Renal outcomes** | | | | |
| Acute kidney injury (AKI), n (%) | 64 (68.1) | 68 (72.3) | 0.81 (0.43–1.53) | 0.517 |
| **AKI Stage distribution, n (%)** | | | | |
| No AKI | 30 (31.9) | 27 (28.7) | −0.14 (−0.40–0.14) | 0.294 |
| Stage 1 | 41 (43.6) | 37 (39.4) | | |
| Stage 2 | 15 (16) | 20 (21.3) | | |
| Stage 3 | 8 (8.5) | 10 (10.6) | | |
| LOS (days), median [IQR] | 29.0 [14.0–53.5] | 22.5 [14.0–39.5] | 0.14 (−0.09–0.35) | 0.261 |

Continuous variables are presented as mean ± standard deviation or median [interquartile range]. Categorical variables are presented as frequencies (percentages). The paired t-test or the Wilcoxon signed-rank test was used for continuous or ordinal outcomes. The McNemar's test, McNemar's exact test, or conditional logistic regression was used for binary outcomes. Non-parametric tests were reported if regression did not converge; descriptive statistics were reported if statistical comparison was not feasible. *p < 0.05 indicates statistical significance. §Effect size or p-value not estimable due to sparse data or zero-cell counts. †Firth's penalised logistic regression was applied at the 90-day time point to obtain an estimate. Effect sizes were reported as odds ratios for binary outcomes, rank-biserial r for ordinal or continuous outcomes. Abbreviations: CI, confidence interval; EAD, early allograft dysfunction; LOS, length of stay; MT, massive transfusion; PNF, primary non-function; UMT, ultramassive transfusion.

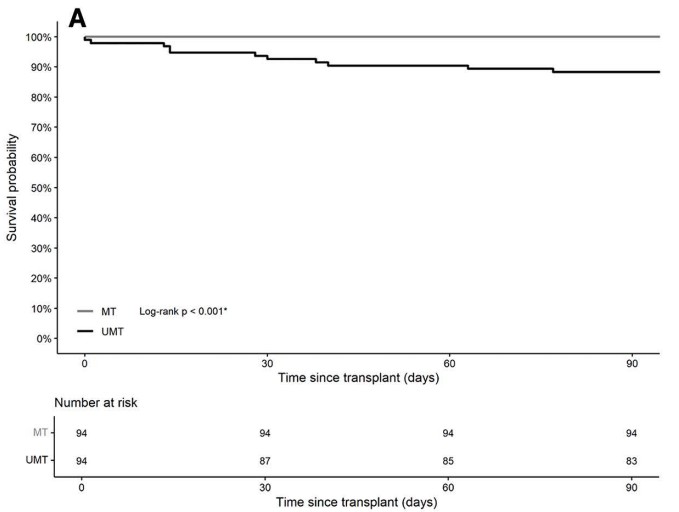
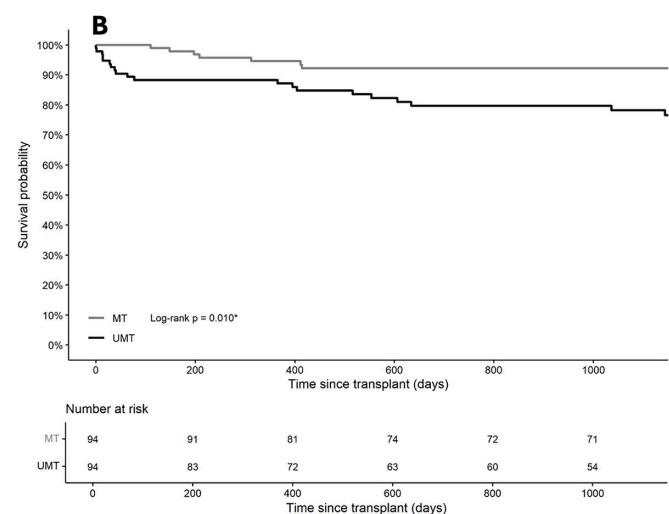

**Fig 3. Early and intermediate patient survival in the propensity-matched cohort.** Kaplan-Meier curves displaying survival probability in the propensity-matched cohort, comparing ultramassive transfusion (≥20 L intraoperative fluids) to massive transfusion (10–20 L intraoperative fluids). Log-rank *p*-values are provided in each panel. (A) 90-day patient survival: Survival was significantly lower in the UMT group (log-rank *p*<0.001). (B) 3-year patient survival (3 years): Survival remained meaningfully lower in the UMT group (log-rank *p*=0.010).

**Table 3. Conditional Cox proportional hazards regression for patient and graft survival in the propensity-matched cohort.**

| Outcome | HR (95% CI) | *p* | PH Global *p*-value |
|---|---|---|---|
| **Patient survival** | | | |
| 90-day survival§ | – | – | – |
| 90-day survival† | 24.30 (3.17–3,119) | <0.001* | – |
| 3-year survival | 2.57 (1.07–6.16) | 0.034* | 0.279 |
| Overall survival | 2.56 (1.18–5.52) | 0.017* | 0.547 |
| **Graft survival** | | | |
| 90-day graft survival | 3.00 (0.61–14.86) | 0.178 | 1.000 |
| 3-year graft survival | 4.50 (0.97–20.83) | 0.054 | 0.445 |
| Overall graft survival | 4.50 (0.97–20.83) | 0.054 | 0.445 |

Conditional Cox proportional hazards regression models stratified by matched pair were used to estimate hazard ratios (HR), 95% confidence intervals (CI), and p-values comparing ultramassive transfusion (≥20 L intraoperative fluids) with massive transfusion (10–20 L) across 90-day, 3-year, and overall follow-up periods. The proportional hazards assumption was verified using Schoenfeld residuals and held for all reported models (PH Global p>0.05). §The conditional Cox model for 90-day patient survival did not converge due to sparse events. †Firth's penalised Cox regression was applied at the 90-day time point to obtain an estimate; the proportional hazards test was deferred (N/A). *p<0.05 indicates statistical significance. Abbreviations: HR, hazard ratio; CI, confidence interval; PH, proportional hazards; N/A, not applicable.

## Sensitivity analyses

Sensitivity Analysis I compared UMT recipients (*n*=94) to a broader control group of all non-UMT recipients (<20 L of intraoperative fluids; *n*=282) using a nearest neighbour 1:3 matching strategy, which reinforced the primary conclusions with increased statistical power. All covariates achieved adequate balance after matching, although partial graft status was borderline (SMD=0.101) (S6 Table). Patient survival remained significant at all time points under conditional logistic regression (all *p*≤0.001; S7 Table), KM curves (all *p*<0.001; S4 Fig) and univariate conditional Cox regression (all

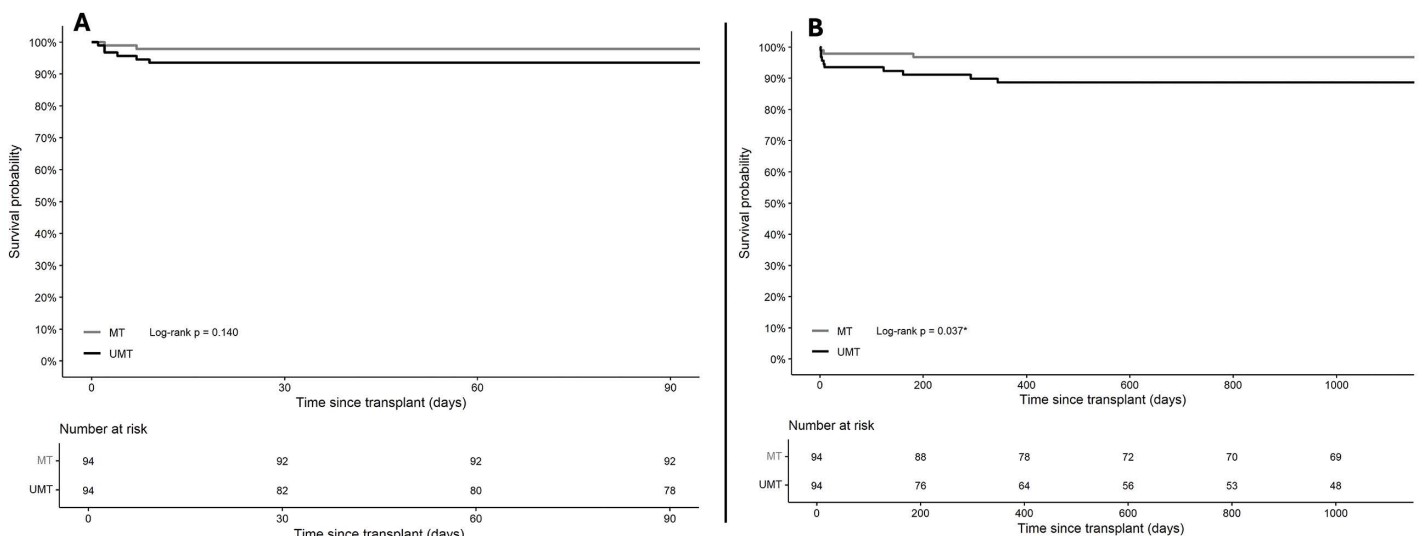

**Fig 4. Early and intermediate graft survival in the propensity-matched cohort.** Kaplan-Meier curves displaying survival probability in the propensity-matched cohort, comparing ultramassive transfusion (≥20 L intraoperative fluids) to massive transfusion (10–20 L intraoperative fluids). Log-rank *p*-values are provided in each panel. (A) 90-day graft survival: The difference was not statistically significant (log-rank *p* = 0.140). (B) 3-year graft survival: Graft survival was significantly lower in the UMT group (log-rank *p* = 0.037).

*p* < 0.001; S8 Table). UMT recipients had increased odds of suffering PNF, EAD, 3-year graft loss and prolonged LOS (all *p* < 0.05)as well as increased hazard of 3-year and overall graft loss (all p < 0.001; S8 Table). Moreover, there was a survival difference in the 3-year (log-rank *p* = 0.012) and overall graft loss (log-rank *p* = 0.030) KM curves (S4 Fig). These findings provide further support for the tenuous associations observed in the primary and unmatched analyses. However, this graft loss effect was not corroborated at 90 days in this sensitivity analysis. Overall, secondary and exploratory outcomes were generally consistent with the primary findings, with no robust new signals detected.

Sensitivity Analysis II examined whether the UMT-mortality association detected in the primary analysis was specific to the composite fluid definition by redefining the exposure using units of intraoperative pRBC transfused only. The conventional UMT threshold of ≥20 units pRBC was evaluated initially but yielded an insufficient number of patients for further analysis. The analysis was conducted using a UMT threshold of ≥15 units pRBC (*n* = 48) compared with MT (10–15 units; *n* = 48). After 1:1 nearest-neighbour matching with a caliper of 0.2 standard deviations, two covariates remained marginally above the conventional balance threshold after matching (cold ischaemia time, SMD = 0.129; partial graft, SMD = 0.157; S9 Table). No statistically significant associations were observed between UMT and any outcome in the matched sensitivity cohort (S10 Table). The hazard of death trended toward worse outcomes for UMT recipients at 90 days (HR 2.00, 95% CI 0.50–8.00, *p* = 0.327), 3 years and overall (HR 1.13, 95% CI 0.43–2.92, *p* = 0.808), but did not reach statistical significance at any time point (S11 Table). KM curves (S5 Fig) showed no significant differences between groups across any time point for patient survival (90-day log-rank *p* = 0.307; 3-year log-rank *p* = 0.414; and overall log-rank *p* = 0.585) and graft survival (90-day log-rank *p* = 0.420; 3-year log-rank *p* = 0.974; and overall log-rank *p* = 0.871). Whilst not significant, the directional consistency of worse patient survival for UMT recipients corroborates the primary analysis, possibly suggesting this sensitivity analysis was underpowered to reveal a mortality signal.

Sensitivity Analysis III assessed whether secular changes in clinical practice over the 15-year study period could account for the observed UMT-mortality association, using the median transplant date (29 October 2018) as the era cut point to create an early era (34 UMT, 53 MT in the matched cohort) and a contemporary era (60 UMT, 41 MT). Era-adjusted conditional Cox regression analyses demonstrated that the UMT-mortality association persisted after

accounting for transplant period, with hazard ratios virtually unchanged from the primary analysis for both 3-year mortality (HR 2.55, 95% CI 1.06–6.11, *p* = 0.036) and overall mortality (HR 2.54, 95% CI 1.17–5.49, *p* = 0.018; S12 Table). Testing for interaction between transplant era and UMT status revealed no evidence that the effect of UMT on mortality differed between eras (3-year: *p* = 0.350; overall: *p* = 0.266), suggesting the effect of UMT on survival was consistent throughout the study period. At the 90-day time point, complete separation persisted within both eras such that the era-adjusted Cox model did not converge. Overall, these findings suggest that the UMT-mortality association is not attributable to secular changes in transplant practice.

## Discussion

### Key findings

In this propensity-matched cohort study of adult OLT recipients, UMT was consistently associated with significantly worse patient survival compared to MT. The robustness of this mortality signal across the early (90 days), intermediate (3 years), and long-term periods, as well as its persistence in the unmatched cohort and across two of the three sensitivity analyses, supports a consistent association between UMT and worse patient survival.The pRBC-centred sensitivity analysis produced directionally consistent but non-significant results, most likely reflecting insufficient statistical power rather than the absence of effect. The association between UMT and graft survival, as well as other exploratory secondary outcomes (including PNF, EAD, thrombotic complications, AKI, and hospital LOS), was less consistent or not significant, likely due to limitations in statistical power.

### Relationship to the existing literature

Our primary finding aligns with existing evidence demonstrating a strong correlation between transfusion volume and patient survival in OLT [8,10,14]. Specifically, prior work in OLT revealed higher mortality in a propensity-matched comparison of MT and non-MT recipients [14]. Another study revealed a dose-dependent effect on mortality in OLT patients who met the MT threshold [10]. This is further supported by studies identifying specific high-volume thresholds, such as receipt of more than 28 units of pRBCs, as a risk factor after transplantation [8].

Our study extends this work by directly comparing UMT against MT in a propensity-matched cohort, suggesting that escalating volumes beyond the conventional MT threshold define a clinically distinct and high-risk clinical phenotype in OLT. Interestingly, patient survival was relatively satisfactory despite the extreme nature of the intervention; the matched cohort had an overall 90-day survival rate of 88.3%, consistent with a UMT study that reported a survival rate of 81.6% during the same perioperative window for volumes exceeding 50 units of pRBCs [16]. Such relatively high survival rates may suggest that high-volume transfusions are not futile, at least in the context of OLT, which is a point of contention in the literature [11,17]. The survival deficit associated with UMT persisted and was robust even after adjustment for baseline confounding and sensitivity analyses, suggesting that UMT exposure is a clinically high-risk phenotype. The mechanisms that may underlie this association are likely multifactorial, involving the sequelae of massive resuscitation, such as haemo-dilutional coagulopathy, acid-base derangements, hypothermia, and endothelial injury stemming from the underlying procedural complexity.While a recent UMT study found that 30-day mortality was not correlated with escalating intraoperative transfusion volumes [24], cross-study comparisons should be interpreted cautiously due to the heterogeneous definitions of MT and UMT as well as different follow-up windows. Notwithstanding, the evidence overarchingly demonstrates that transfusion volume is a clinically meaningful prognostic indicator in OLT.

The association between UMT and graft outcomes was less robust than for patient survival. Our mortality-censored Cox model showed only a borderline association with graft loss at 3 years. However, the elevated hazard became statistically significant at 3 years and overall, when the analysis was performed in the higher-powered sensitivity cohort comparing UMT with all non-UMT recipients (<20 L), suggesting a possible pattern of increasing risk across transfusion thresholds.

Similarly, a higher frequency of rare early outcomes, including PNF and certain thrombotic complications, was observed among UMT recipients. The small number of events for these secondary outcomes in the primary matched cohort likely contributed to the wide confidence intervals and statistical non-significance. The estimates of these secondary outcomes should be interpreted cautiously, as they likely reflect limited statistical power rather than robust evidence of no association, particularly given that our sensitivity analysis with a broader comparator group did find significant associations for PNF and EAD. However, these associations were not corroborated in other sensitivity analyses. The association with thrombotic complications was non-significant in contrast to expectations based on recent literature [30].

Other exploratory secondary outcomes were not significant, including the incidence and severity of postoperative AKI. The significant association between UMT and EAD observed in the unmatched cohort was attenuated after matching, suggesting the initial correlation was predominantly driven by baseline confounding, rather than exposure to extreme transfusion volumes. This is consistent with another study that found an association between EAD and pRBC units in univariate but not multivariate analysis [31]. However, the significant association between UMT and EAD observed in Sensitivity Analysis I, which used a larger control group, suggests that the attenuation in the primary matched cohort may also partly reflect limited statistical power rather than confounding alone. Our primary analysis did not identify a significant association with prolonged hospital LOS, replicating the findings of an MT study of OLT patients [14], but contrasting with a recent liver transplant UMT study [24], which reported a significant association with LOS in intensive care.

## Clinical implications

Our findings suggest that UMT recipients are a distinct, high-risk clinical subpopulation of OLT patients, possibly independent of pre-transplant morbidity. Identifying patients who receive ≥ 20 L of intraoperative fluids during their transplant should prompt heightened perioperative vigilance and judicious resource allocation. Our findings inform anticipatory measures such as early engagement of critical care services, targeted postoperative surveillance in the early (90 days) and intermediate (3 years) periods, and the need for OLT-specific massive transfusion or UMT protocols. The persistent survival difference observed in the long term suggests this subpopulation of patients may warrant long-term postoperative follow-up, potentially including lifestyle or pharmacological interventions, to recognise transfusion-related sequelae and improve patient survival in the early, intermediate and long-term periods.

## Strengths and limitations

This study's primary strength lies in leveraging PSM to mitigate confounding and create comparable UMT and MT groups. This methodological approach provides robust estimates of the relationship between extreme transfusion volume and patient outcomes. The confidence in our primary finding is supported by its consistency and robustness across unmatched, matched and sensitivity analyses using alternative matching strategies.

This study has several limitations. The retrospective design introduces the possibility of residual confounding, as PSM can only account for measured variables included in the data. Critically, the propensity score model was constructed using preoperative variables only. Key intraoperative factors that are likely to influence both transfusion volume and patient outcomes [14,16], including estimated blood loss, vasopressor requirement and surgical complications, were not captured in the transplant registry data. Thus, UMT may reflect intraoperative severity rather than independently cause worse outcomes. Patients receiving such extreme resuscitation volumes are plausibly those who also receive technically challenging surgeries or suffer catastrophic blood loss due to other intraoperative factors. This confounding by indication cannot be fully addressed by the study design, and the observed association between UMT and mortality should be interpreted with caution. Survivor bias is also a potential concern; patients who survived long enough to receive extreme transfusion volumes and to be captured in the registry may represent a physiologically resilient subgroup, potentially leading to underestimation of the mortality risk associated with UMT.

Regarding temporal confounding, a prespecified time-period sensitivity analysis demonstrated that the UMT-mortality association persisted after adjusting for transplant era and was consistent in magnitude across the early (2009–2018) and contemporary (2018–2023) periods, with no significant era interaction detected. This finding suggests that secular changes in clinical practice over the 15-year study period are unlikely to account for the observed association. Graft survival and secondary outcomes were not included in the time-period sensitivity analysis, as no significant and consistent association were observed for these endpoints in the primary analysis or other sensitivity analyses.

The pRBC sensitivity analysis was limited by the small number of patients classifiable under the conventional threshold of ≥20 units; the lower threshold of ≥15 units used may not fully capture the extreme transfusion burden intended by the UMT designation. This threshold is uncommon in prior literature [11], further limiting the interpretability of these results. The non-significant findings at this threshold cannot exclude a pRBC-specific effect at higher transfusion volumes, especially given that the direction of effect was consistent with the primary analysis across all time points.

While our sample size was substantial for a single-centre OLT study focused on UMT, it is considerably smaller than those in multicentre studies of high-volume transfusions in trauma [23,32]. Our study was likely underpowered to detect robust differences for rarer outcomes, which predisposed instability and insignificance in our secondary exploratory outcomes, as demonstrated in the tenuous relationship between transfusion volume and graft loss, where a significant effect was observed at 3 years but not at other perioperative windows. The conditional Cox models for 3-year and overall graft survival in the primary analysis and patient survival in the pRBC-centred sensitivity analysis produced identical estimates, likely reflecting sparse event counts and underpowered analyses. Generalisability is limited as this study reflects practice at a single Australian quaternary referral centre with mature blood bank infrastructure. Differences in institutional resuscitation protocols, blood bank infrastructure, and resource constraints can impact patient outcomes [10,12] and, by extension, the findings reported in our study.

The inconsistent definitions of UMT and MT in the literature hinder cross-study comparisons, such as ≥ 20 units of pRBC in 24 hours for UMT in trauma [11], ≥ 6 units of pRBC in 24 hours for MT in OLT [10], or even ≥ 50 units of pRBC in 24 hours for 'supermassive' transfusions in OLT [16]. Most published definitions use blood product units alone, whereas this study defines transfusion burden as the total fluid volume, including crystalloids and colloids. While this definition captures the full burden of resuscitation and is consistent with prior work at our institution [24], it limits direct comparison with studies using pRBC-only thresholds. Thus, our findings may reflect differences in fluid management strategy rather than transfusion burden, introducing uncertainty when interpreting the significance of our findings. The attenuation of the mortality signal observed in our pRBC-centred sensitivity analysis is consistent with this interpretation, suggesting that the total resuscitative burden, including crystalloids and colloids, may be a more complete and clinically meaningful measure of intraoperative physiological stress than pRBC transfusion volume alone.

Finally, in interpreting 90-day mortality estimates in Tables 2 and 3, it is important to note that complete separation and sparse early events required the use of Firth's penalised logistic and Cox models, resulting in very large effect estimates with extremely wide confidence intervals (for example, HR 24.30, 95% CI 3.17–3,119), which we consider descriptive support for an association rather than a reliable measure of a 26-fold increase in risk.

## Implications for future research

The consistent association between UMT and mortality across all analyses identifies UMT as a high-risk, potentially modifiable exposure that warrants focused investigation. Future studies should aim to elucidate the mechanisms underlying this persistent survival deficit, including transfusion-related complications, endothelial injury, and unmeasured intraoperative factors, and to evaluate targeted strategies to mitigate risk, such as optimised component therapy, haemostatic interventions, and structured postoperative surveillance.

To strengthen causal inference, multicentre collaborations incorporating detailed intraoperative and postoperative data are needed to validate these findings and improve statistical power for rare outcomes. Future studies examining high-volume transfusion in OLT should consider reporting total fluid volumes alongside pRBC units to enable more

comprehensive cross-study comparisons and to clarify whether the mortality signal is attributable to transfused blood products specifically or to the broader haemodynamic and haemodilutional consequences of extreme resuscitation. Finally, standardising definitions of massive and ultramassive transfusion in OLT is essential to harmonise research and facilitate cross-study comparisons, particularly in the non-trauma setting.

## Conclusion

This study provides novel evidence that UMT is associated with significantly reduced patient survival in adult OLT recipients compared with MT. The consistent and persistent mortality signal observed across matched and unmatched cohorts, as well as sensitivity analyses, is consistent with a dose-dependent pattern of association, wherein extreme intraoperative fluid exposure may characterise a distinct, high-risk clinical phenotype. A pRBC-centred sensitivity analysis produced directionally consistent but non-significant results, most likely reflecting insufficient statistical power at the available threshold rather than the absence of effect. However, due to the limitations of our study, our findings cannot establish a causal relationship; UMT may reflect intraoperative severity rather than independently drive mortality. Nonetheless, our findings suggest that risk mitigation strategies should extend beyond immediate perioperative haemostatic optimisation to include heightened clinical vigilance and tailored long-term surveillance for UMT recipients. Future prospective, multicentre studies are needed to clarify underlying mechanisms, refine transfusion thresholds, and identify modifiable factors that may improve both short- and long-term outcomes in this vulnerable population.

## Supporting information

**S1 Fig. Unmatched analysis: Love plots of covariate balance across multiple propensity score matching strategies.**
(PDF)

**S2 Fig. Primary analysis: Long-term survival comparisons in the propensity-matched cohort.**
(PDF)

**S3 Fig. Unmatched analysis: Early, intermediate and long-term survival comparison in the unmatched cohort.**
(PDF)

**S4 Fig. Sensitivity analysis I (expanded comparator): Matched survival curves.**
(PDF)

**S5 Fig. Sensitivity analysis II (pRBC definition): Matched survival curves.**
(PDF)

**S1 Table. Unmatched analysis: Summary of missing data by variable.**
(PDF)

**S2 Table. Primary analysis: Evaluation of the covariate balance across multiple propensity score matching strategies.**
(PDF)

**S3 Table. Primary analysis: Complete baseline recipient, donor and intraoperative characteristics before and after matching.**
(PDF)

**S4 Table. Unmatched analysis: Perioperative and long-term outcomes.**
(PDF)

**S5 Table. Unmatched analysis: Conditional Cox proportional hazards regression for patient and graft survival.**
(PDF)

**S6 Table. Sensitivity analysis I (expanded comparator): Evaluation of covariate balance.**
(PDF)

**S7 Table. Sensitivity analysis I (expanded comparator): Perioperative and long-term outcomes.**
(PDF)

**S8 Table. Sensitivity analysis I (expanded comparator): Conditional Cox proportional hazards regression for patient and graft survival in the matched sensitivity cohort.**
(PDF)

**S9 Table. Sensitivity analysis II (pRBC exposure): Covariate balance assessment.**
(PDF)

**S10 Table. Sensitivity analysis II (pRBC exposure): Perioperative and long-term outcomes in the matched sensitivity cohort.**
(PDF)

**S11 Table. Sensitivity analysis II (pRBC exposure): Conditional Cox proportional hazards regression for patient and graft survival in the matched sensitivity cohort.**
(PDF)

**S12 Table. Sensitivity analysis III (time period): Conditional Cox proportional hazards regression for patient survival in the matched cohort.**
(PDF)

**S1 File. De-identified Database.**
(XLSX)

## Ackowledgements

Nil.

## Author contributions

**Conceptualization:** Laurence Weinberg.

**Data curation:** Rebecca Caragata, Marcos V. Perini, Anoop N Koshy, Laurence Weinberg.

**Formal analysis:** Zac Tran, Jemin Suh, Dong-Kyu Lee.

**Investigation:** Zac Tran, Nattaya Raykateeraroj, Jemin Suh, Michael Fink, Rebecca Caragata, Marcos V. Perini, Anoop N Koshy, Dong-Kyu Lee, Laurence Weinberg.

**Methodology:** Zac Tran, Nattaya Raykateeraroj, Jemin Suh, Michael Fink, Laurence Weinberg.

**Project administration:** Laurence Weinberg.

**Resources:** Laurence Weinberg.

**Supervision:** Dong-Kyu Lee, Laurence Weinberg.

**Validation:** Zac Tran, Laurence Weinberg.

**Visualization:** Zac Tran, Laurence Weinberg.

Writing – original draft: Zac Tran, Nattaya Raykateeraroj, Laurence Weinberg.

Writing – review & editing: Zac Tran, Nattaya Raykateeraroj, Jemin Suh, Jordan Ismail, Michael Fink, Rebecca Caragata, Marcos V. Perini, Anoop N Koshy, Dong-Kyu Lee, Laurence Weinberg.

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
