## [Decision Letter · Decision Letter 0]

10 Mar 2026

PONE-D-26-07916Effects of massive transfusion (10-20 litres) versus ultramassive transfusion (≥20 litres) on mortality in adult liver transplant recipients: A propensity-score matched studyPLOS One

Dear Dr. Weinberg,

Thank you for submitting your manuscript to PLOS ONE. After careful consideration, we feel that it has merit but does not fully meet PLOS ONE’s publication criteria as it currently stands. Therefore, we invite you to submit a revised version of the manuscript that addresses the points raised during the review process.

We look forward to receiving your revised manuscript.

Kind regards,

Pavel Strnad

Academic Editor

PLOS One

Journal Requirements:

2. We note that there is identifying data in the Supporting Information file < Deidentified Database_PLOS One_13Feb2026>. Due to the inclusion of these potentially identifying data, we have removed this file from your file inventory. Prior to sharing human research participant data, authors should consult with an ethics committee to ensure data are shared in accordance with participant consent and all applicable local laws.

-Location data

Reviewers' comments:

Reviewer's Responses to Questions

**Comments to the Author**

1. Is the manuscript technically sound, and do the data support the conclusions?

Reviewer #1: Yes

Reviewer #2: Yes

2. Has the statistical analysis been performed appropriately and rigorously? 

Reviewer #1: Yes

Reviewer #2: Yes

3. Have the authors made all data underlying the findings in their manuscript fully available?

Reviewer #1: Yes

Reviewer #2: Yes

4. Is the manuscript presented in an intelligible fashion and written in standard English?

Reviewer #1: Yes

Reviewer #2: Yes

5. Review Comments to the Author

Reviewer #1: This retrospective single-centre study evaluates the association between ultramassive transfusion (≥20 L total fluids) and mortality in adult liver transplant recipients using propensity score matching. The topic is clinically relevant and the dataset represents a valuable institutional experience. The manuscript addresses an important clinical question regarding the impact of very high intraoperative transfusion volumes in liver transplantation. However, several methodological aspects limit the interpretation of the findings and should be clarified or more thoroughly discussed.

Major Comments:

The main limitation of the study is that the analysis does not establish a causal relationship between transfusion volume and mortality. Patients receiving ultramassive transfusion are very likely those experiencing the most severe intraoperative complications (e.g., major bleeding, technically difficult surgery, graft issues), which are themselves strong predictors of mortality. Therefore, ultramassive transfusion is likely a marker of intraoperative severity rather than an independent cause of worse outcomes. Although propensity score matching was performed, the model appears to include mainly preoperative variables and donor characteristics, while important intraoperative severity markers such as blood loss, operative duration, vasopressor requirement, or intraoperative complications were not included. This creates a high risk of residual confounding and confounding by indication. This issue should ideally be addressed through additional clarification or analyses if possible, or at least more clearly acknowledged and discussed as a key limitation of the study.

A second concern relates to the definition of the exposure variable. Massive and ultramassive transfusion are defined using total intraoperative fluid volume, including crystalloids and colloids. This differs from most definitions in the literature, which typically rely on the number of blood product units. Including non-blood fluids may reflect differences in anaesthetic fluid management strategies rather than transfusion burden alone, which complicates interpretation of the results. The rationale for this definition should be explained more clearly. If this approach cannot be further justified or explored analytically, its implications for interpretation should be explicitly acknowledged and discussed as a limitation.

Minor Comments:

There are several minor issues that should be addressed during revision. Some spelling and typographical errors are present in the manuscript and should be corrected. In addition, several internal references appear to be incorrect or unresolved (e.g., “Error! Reference source not found.”), suggesting formatting problems in the document. These should be carefully checked and corrected before publication.

Reviewer #2: This propensity-matched cohort study addresses a clinically important question in orthotopic liver transplantation (OLT). The primary finding is credible and the study makes a meaningful contribution. Several methodological issues require revision before publication:

The mortality signal is robust: its consistency across the matched cohort, unmatched cohort, and 1:3 sensitivity analysis (at 90 days, 3 years, and overall follow-up) substantially strengthens confidence in the finding. The PSM with 11 preoperative covariates achieves excellent balance as shown by the Love plots. The 15-year single-centre dataset from a high-volume transplant unit ensures data consistency and sufficient case volume for a rare exposure. The transparent handling of underpowered secondary outcomes reflects appropriate statistical discipline.

Major Concerns

1. Confounding vs. Independent Risk — Causal Language

The PSM adjusts for preoperative variables only. Critical intraoperative factors (surgical complexity, operative duration, and bleeding aetiology) remain unmeasured. UMT may therefore reflect intraoperative factors rather than independently drive mortality, a distinction the design cannot resolve. The attenuation of the EAD association after matching illustrates this mechanism directly. Causal language in the abstract and conclusion (“independently associated,” “linked to”) should be replaced throughout with “associated with,” and the conclusion should explicitly acknowledge that causal inference is not supported.

2. Composite Fluid Definition

Combining crystalloids, colloids, and blood products into a single threshold conflates physiologically distinct entities. A sensitivity analysis using a pRBC-centred definition would strengthen cross-study comparability and help disentangle haemodilution from transfusion-related immunomodulation.

3. 90-Day Mortality: Missing Effect Estimate

The 90-day mortality difference (11.7% vs. 0%) is the most striking finding, yet no effect size is reported because the Cox model did not converge. A Firth-corrected logistic regression estimate or exact confidence interval should improve the manuscript.

Minor Concerns

Table 1 contains a likely typographical error (“27.4.0 L”). A time-period sensitivity analysis would help assess whether secular changes in practice over the 15-year study window confound the association.

6. PLOS authors have the option to publish the peer review history of their article (what does this mean?). If published, this will include your full peer review and any attached files.

Reviewer #1: No

Reviewer #2: No

---

## [Author Response · Author response to Decision Letter 1]

6 Apr 2026

RESPONSE TO REVIEWERS

Professor Pavel Strnad

Academic Editor

PLOS One

PONE-D-26-07916

Title: Effects of massive transfusion (10-20 litres) versus ultramassive transfusion (≥20 litres) on mortality in adult liver transplant recipients: A propensity-score matched study

Dear Pavel Strnad

My coauthors and I would like to thank you and the expert reviewers for taking the time to provide a very constructive and thoughtful review of our manuscript.

As requested, we have included the following items with our resubmission:

1. A point-by-point response letter (to follow) that addresses each issue raised by the Academic Editor and the Reviewers. I have uploaded this letter as a separate file labelled “Response to Reviewers.”

2. A marked-up copy of the manuscript that highlights changes made to the original version. This has been uploaded as a separate file labelled “Revised Manuscript with Tracked Changes highlighted in red.”

3. An unmarked version of our revised paper without tracked changes. This has also been uploaded as a separate file labelled “Manuscript.”

Editorial / General Points

1. PLOS ONE style and file naming

Authors’ response: We thank the Editors for this important reminder and have carefully reviewed the manuscript and all associated files to ensure compliance with PLOS ONE’s formatting and file-naming requirements. All manuscript, figure, and supplementary files have now been renamed and formatted in accordance with the journal’s guidelines.

2. Anonymisation of supporting data

Authors’ response: We are grateful for the Editors’ vigilance regarding patient confidentiality and data governance. Following your advice, we have reviewed the original dataset in detail, removed all directly identifying fields, and ensured that no potentially identifying information remains (including hidden spreadsheet columns). The re-uploaded data file now contains only fully anonymised variables, consistent with the consent framework approved by our Human Research Ethics Committee and applicable local regulations.

3. Citation of suggested references

Authors’ response: We thank the Editors and reviewers for their thoughtful reference suggestions. We have systematically reviewed the recommended publications and have cited those that are directly relevant to our study question, methods, or interpretation, while avoiding citation purely for completeness. Where suggested articles were not clearly applicable to orthotopic liver transplantation (OLT) or to ultramassive transfusion specifically, we have acknowledged the broader literature without adding references that might distract from the OLT focus of the manuscript.

Reviewer 1

We sincerely thank Reviewer 1 for the careful reading of our manuscript and for their insightful comments, which have substantially improved the clarity and interpretability of our work. We agree that each of the points raised is important and have addressed them in detail as follows.

4. Causality, intraoperative severity, and residual confounding

Reviewer’s comment: “The main limitation of the study is that the analysis does not establish a causal relationship between transfusion volume and mortality. Patients receiving ultramassive transfusion are very likely those experiencing the most severe intraoperative complications (e.g., major bleeding, technically difficult surgery, graft issues), which are themselves strong predictors of mortality. Therefore, ultramassive transfusion is likely a marker of intraoperative severity rather than an independent cause of worse outcomes. Although propensity score matching was performed, the model appears to include mainly preoperative variables and donor characteristics, while important intraoperative severity markers such as blood loss, operative duration, vasopressor requirement, or intraoperative complications were not included. This creates a high risk of residual confounding and confounding by indication. This issue should ideally be addressed through additional clarification or analyses if possible, or at least more clearly acknowledged and discussed as a key limitation of the study.”

Authors’ response: We thank the Reviewer for raising this concern. We acknowledge that residual confounding and confounding by indication are inherent limitations of this retrospective propensity-matched study. Moreover, we agree that intraoperative variables, such as the volume of blood loss, surgical complexity and intraoperative complications, are likely important predictors of mortality. Unfortunately, this data was not captured in the transplant registry and was subsequently unavailable for our statistical analysis.

In response to this concern, we attempted to minimise confounding and to test the robustness of the primary findings. The propensity score matching strategy included 11 clinically relevant variables and achieved adequate covariate balance, with standardised mean differences (SMD) below 0.1 after matching (Figure 2; Supplementary Table 2). This balance post-matching demonstrates that the exposure groups in the primary analysis were well-matched on available characteristics. Moreover, this study was conducted at a single quaternary centre specialising in liver transplantation, ensuring internal consistency in clinical practice, which may mitigate residual confounding.

The consistency of the mortality signal across the unmatched and sensitivity analyses suggests that the UMT-mortality association may not be attributable to confounding alone. The association was observed in the unmatched cohort (n = 306), the primary matched cohort (n = 188), and the expanded cohort comparing UMT against all non-UMT (< 20 L intraoperative fluids) recipients (n = 376). Additionally, the mortality signal remained in the time-period sensitivity analysis to detect whether secular changes in clinical practice at our institution contributed to the UMT-mortality association.

Thus, this pattern of consistency suggests that the primary mortality findings are unlikely to be attributable to confounding alone, even in the absence of critical intraoperative variables.

To address this important concern, we have made the following changes:

I. Strengthened limitations section: We have expanded the Strengths and Limitations subsection to explicitly state that “critically, the propensity score model was constructed using preoperative variables only,” and that key intraoperative factors influencing both transfusion volume and outcomes were unavailable. We now explicitly acknowledge confounding by indication, emphasizing that UMT “may reflect intraoperative severity rather than independently cause worse outcomes,” and that the observed association “should be interpreted with caution.”

II. Clarified interpretation in Discussion and Conclusion: We now consistently describe UMT as identifying a “high‐risk” clinical phenotype and carefully avoid implying that UMT is itself causal. The Discussion explicitly notes that patients receiving UMT are plausibly those experiencing greater intraoperative complexity or catastrophic blood loss, and that this distinction cannot be resolved by our study design

III. Survivor bias and unmeasured confounding: We added a paragraph explicitly describing the possibility of survivor bias (patients must survive long enough to receive extreme volumes) and the broader issue of unmeasured confounding, underscoring that propensity score matching cannot adjust for variables that are not measured.

We believe these revisions make the limitations around causality and residual confounding much more transparent and aligned with the reviewer’s concerns.

5. Exposure definition: total intraoperative fluid volume vs blood product units

Reviewer’s comment: “A second concern relates to the definition of the exposure variable. Massive and ultramassive transfusion are defined using total intraoperative fluid volume, including crystalloids and colloids. This differs from most definitions in the literature, which typically rely on the number of blood product units. Including non-blood fluids may reflect differences in anaesthetic fluid management strategies rather than transfusion burden alone, which complicates interpretation of the results. The rationale for this definition should be explained more clearly. If this approach cannot be further justified or explored analytically, its implications for interpretation should be explicitly acknowledged and discussed as a limitation.”

Authors’ response: We thank the Reviewer for raising this concern. This concern was also raised by Reviewer 2. Accordingly, our pRBC-centred sensitivity analysis addresses both Reviewer comments.

We have addressed this in three ways:

I. Expanded justification in Methods: In the “Patient population and definitions” section, we now provide a clearer, a priori rationale for the composite fluid definition. Specifically, we emphasise that:

• The definition was chosen to capture the “total resuscitation burden”, as all fluid components contribute to haemodilution, acid–base disturbance, endothelial injury, and overall physiological stress in OLT.

• Our approach is consistent with prior work at our institution, which also used a total-fluid definition in an OLT UMT cohort.

• Although not universal, these thresholds (10–20 L vs ≥20 L) represent clinically meaningful checkpoints for resource utilisation, and are aligned with local practice triggers for escalation of perioperative vigilance.

II. Explicit acknowledgment as a limitation: In the Strengths and Limitations section, we now explicitly state that our composite fluid definition “limits direct comparison with studies using pRBC-only thresholds” and may partly reflect differences in anaesthetic fluid management strategies rather than transfusion burden alone. We further highlight that “our findings may reflect differences in fluid management strategy rather than transfusion burden, introducing uncertainty when interpreting the significance of our findings.

III. Additional analytic exploration (pRBC-centred sensitivity analysis): To address the reviewer’s suggestion for more analytic exploration, we have added a dedicated pRBC-centred sensitivity analysis (Sensitivity Analysis II) using pRBC units to define MT and UMT (10–14 vs ≥15 units), with full details in the Methods, Results, and Supplementary Tables 9–11. Although this analysis was underpowered at conventional UMT thresholds, it provides a complementary perspective and supports the notion that total fluid volume may be a more complete index of resuscitative burden in our setting.

We believe these additions provide a clearer justification of our exposure definition and appropriately frame it as both a strength (comprehensive resuscitation measure) and a limitation (comparability with pRBC-based definitions).

6. Spelling, typographical errors, and internal referencing issues

Reviewer’s comment: “There are several minor issues that should be addressed during revision. Some spelling and typographical errors are present in the manuscript and should be corrected. In addition, several internal references appear to be incorrect or unresolved (e.g., “Error! Reference source not found.”), suggesting formatting problems in the document. These should be carefully checked and corrected before publication.”

Authors’ response: We appreciate the reviewer drawing attention to these issues, which have now been addressed systematically. Specifically:

• We performed a comprehensive proofread of the manuscript to correct spelling, grammar, and typographical errors, including those in tables and figure legends.

• All internal referencing errors (e.g. “Error! Reference source not found.”) arising from word-processor cross-references have been identified and corrected so that all tables, figures, and supplementary materials are now referenced correctly and consistently.

• We also standardised terminology (e.g., “ultramassive transfusion,” “massive transfusion,” and abbreviations) throughout the text for readability and consistency.

Reviewer 2

We are very grateful to Reviewer 2 for their thoughtful and constructive review, as well as the positive assessment of the study’s contribution and statistical approach. We agree that each of the major and minor points raised is important and have revised the manuscript accordingly.

7. Overall assessment of robustness and contribution.

Reviewer’s comment: “The PSM adjusts for preoperative variables only. Critical intraoperative factors (surgical complexity, operative duration, and bleeding aetiology) remain unmeasured. UMT may therefore reflect intraoperative factors rather than independently drive mortality, a distinction the design cannot resolve. The attenuation of the EAD association after matching illustrates this mechanism directly.

Authors’ response: We thank the reviewer for these encouraging comments regarding the robustness of the mortality signal, the quality of the propensity score matching, and the transparent handling of secondary outcomes. This comment overlaps with Reviewer 1. We agree that the study design cannot definitively answer whether UMT independently drives mortality or whether it may be a proxy for intraoperative severity. A more detailed response to confounding factors is provided in response to Reviewer 1 Question 1 (see above) and applies here as well.

We have ensured that the manuscript clearly reports:

I. The excellent covariate balance achieved by the 1:1 optimal matching strategy, supported by Love plots and detailed standardised mean differences in Supplementary Tables.

II. The consistency of the mortality association across unmatched analyses and multiple sensitivity analyses, including the expanded comparator (UMT vs all non-UMT recipients) and the pRBC-centred analysis.

III. The “underpowered nature” of certain secondary and exploratory endpoints (e.g., graft loss, rare thrombotic events), with cautious interpretation explicitly noted in both Results and Discussion.

We appreciate the reviewer’s recognition of these aspects and have retained this transparent framing in the revised manuscript.

8. Confounding vs independent risk and causal language

Reviewer’s comment: “Causal language in the abstract and conclusion (“independently associated,” “linked to”) should be replaced throughout with “associated with,” and the conclusion should explicitly acknowledge that causal inference is not supported.”

Authors’ response: We agree entirely with the reviewer that our design does not support causal inference and that the language should clearly reflect an “associational”, rather than causal, interpretation.

Accordingly, we have made the following changes:

I. Abstract:

• Replaced any phrases implying causality (e.g. “independently associated,” “linked to”) with the more neutral “associated with.”

• Added an explicit statement in the Abstract’s Conclusion that “our study does not permit causal inference and the observed association may reflect intraoperative severity rather than a direct effect of transfusion volume.”

II. Main text and Conclusion:

• Throughout the Introduction, Discussion, and Conclusion, we systematically replaced causal-sounding terminology with “associated with,” “identifies a high-risk phenotype,” or “correlated with,” as appropriate.

• The Conclusion now explicitly states that “due to the limitations of our study, our findings cannot establish a causal relationship; UMT may reflect intraoperative severity rather than independently drive mortality.”

III. Limitations:

• The Strengths and Limitations section now prominently highlights that intraoperative factors were not included in the propensity model and that residual confounding and confounding by indication are likely.

We believe these revisions fully address the reviewer’s concern and provide a transparent, appropriately cautious interpretation of our findings.

9. Composite fluid definition and additional pRBC-centred sensitivity analysis

Reviewer’s comment: “Combining crystalloids, colloids, and blood products into a single threshold conflates physiologically distinct entities. A sensitivity analysis using a pRBC-centred definition would str

---

## [Decision Letter · Decision Letter 1]

6 May 2026

Effects of massive transfusion (10-20 litres) versus ultramassive transfusion (≥20 litres) on mortality in adult liver transplant recipients: A propensity-score matched study

PONE-D-26-07916R1

Dear Dr. Weinberg,

We’re pleased to inform you that your manuscript has been judged scientifically suitable for publication and will be formally accepted for publication once it meets all outstanding technical requirements.

Kind regards,

Pavel Strnad

Academic Editor

PLOS One

Additional Editor Comments (optional): Congratulations to the nice manuscript!

Reviewers' comments:

Reviewer's Responses to Questions

**Comments to the Author**

1. If the authors have adequately addressed your comments raised in a previous round of review and you feel that this manuscript is now acceptable for publication, you may indicate that here to bypass the “Comments to the Author” section, enter your conflict of interest statement in the “Confidential to Editor” section, and submit your "Accept" recommendation.

Reviewer #1: All comments have been addressed

2. Is the manuscript technically sound, and do the data support the conclusions?

Reviewer #1: Yes

3. Has the statistical analysis been performed appropriately and rigorously? 

Reviewer #1: Yes

4. Have the authors made all data underlying the findings in their manuscript fully available?

Reviewer #1: Yes

5. Is the manuscript presented in an intelligible fashion and written in standard English?

Reviewer #1: Yes

6. Review Comments to the Author

Reviewer #1: Thank you for the opportunity to review the revised version of this manuscript.

The authors have addressed my previous comments in a thoughtful and comprehensive manner. In particular, the limitations regarding causality, residual confounding, and the absence of intraoperative variables are now clearly acknowledged and appropriately discussed. The interpretation of the findings has been revised accordingly, with a more cautious and balanced framing that avoids causal language.

I also appreciate the authors’ efforts to strengthen the methodological transparency, including the addition of a pRBC-based sensitivity analysis and further clarification of the exposure definition. These additions improve both the interpretability and the contextualisation of the results within the existing literature.

While the inherent limitations of the retrospective design remain, these are now explicitly recognised, and the conclusions are appropriately aligned with the available data.

Overall, I believe the manuscript has been substantially improved and now meets an acceptable standard for publication.

7. PLOS authors have the option to publish the peer review history of their article (what does this mean?). If published, this will include your full peer review and any attached files.

Reviewer #1: No

---

## [Editor Report · Acceptance letter]

PONE-D-26-07916R1

PLOS One

Dear Dr. Weinberg,

I'm pleased to inform you that your manuscript has been deemed suitable for publication in PLOS One. Congratulations! Your manuscript is now being handed over to our production team.

Kind regards,

on behalf of

Dr. Pavel Strnad

Academic Editor

PLOS One